# MIDAS: A Benchmarking Multi-Criteria Method for the Identification of Defective Anemometers in Wind Farms

**Arkaitz Rabanal** [1,‡]**, Alain Ulazia** [2,*,‡] **, Gabriel Ibarra-Berastegi** [3,†,‡] **, Jon Sáenz** [4,†,‡] **and Unai Elosegui** [5,‡]

1   University of the Basque Country (UPV/EHU), Otaola 29, 20600 Eibar, Spain; arabanal004@ikasle.ehu.eus
2   Department of NE and Fluid Mechanics, University of the Basque Country (UPV/EHU), Otaola 29, 20600 Eibar, Spain
3   Department of NE and Fluid Mechanics, University of the Basque Country (UPV/EHU), Alda, Urkijo, 48013 Bilbao, Spain; gabriel.ibarra@ehu.eus
4   Department of Applied Physics II, University of the Basque Country (UPV/EHU), B. Sarriena s/n, 48940 Leioa, Spain; jon.saenz@ehu.eus
5   Maxwind Technology, Portuetxe 83, 20018 Donostia, Spain; unai.elosegui@corp.hispavista.com
*   Correspondence: alain.ulazia@ehu.eus
†   BEGIK, Joint Research Unit (UPV/EHU-IEO) Plentziako Itsas Estazioa (PIE), University of the Basque Country (UPV/EHU), Areatza Hiribidea 47, 48620 Plentzia, Spain.
‡   These authors contributed equally to this work.

**Abstract:** A novel multi-criteria methodology for the identification of defective anemometers is shown in this paper with a benchmarking approach: it is called MIDAS: multi-technique identification of defective anemometers. The identification of wrong wind data as provided by malfunctioning devices is very important, because the actual power curve of a wind turbine is conditioned by the quality of its anemometer measurements. Here, we present a novel method applied for the first time to anemometers' data based on the kernel probability density function and the recent reanalysis ERA5. This estimation improves classical unidimensional methods such as the Kolmogorov–Smirnov test, and the use of the global ERA5's wind data as the first benchmarking reference establishes a general method that can be used anywhere. Therefore, adopting ERA5 as the reference, this method is applied bi-dimensionally for the zonal and meridional components of wind, thus checking both components at the same time. This technique allows the identification of defective anemometers, as well as clear identification of the group of anemometers that works properly. After that, other verification techniques were used versus the faultless anemometers (Taylor diagrams, running correlation and *RMSE*, and principal component analysis), and coherent results were obtained for all statistical techniques with respect to the multidimensional method. The developed methodology combines the use of this set of techniques and was able to identify the defective anemometers in a wind farm with 10 anemometers located in Northern Europe in a terrain with forests and woodlands. Nevertheless, this methodology is general-purpose and not site-dependent, and in the future, its performance will be studied in other types of terrain and wind farms.

**Keywords:** wind turbine; anemometer; kernel-based multidimensional probability density function; ERA5 reanalysis

## 1. Introduction

Maintenance is a critical variable in the wind industry in order to reach competitiveness. Failure detection and diagnosis is essential, as is the study of the relation of these faults with energy

production, profitability, costs, and safety. Toward this end, several advances in mathematics and computational techniques are employed in maintenance management: dynamic analysis, probabilistic methods, mathematical optimization techniques, etc. The combination of these techniques enables a multi-criteria diagnosis and decision-making processes for different problems in wind farms [1–4].

In the context of operation and maintenance (O&M), the identification of defective anemometers that are located in wind turbines is important, because they are used for the estimation of the energy produced by the turbine through its actual power curve.

For each wind energy application, the type of instrumentation required varies widely from a simple system containing only one wind speed anemometer/recorder to a very complex system designed to characterize turbulence across the rotor plane. This kind of instrumentation is very important for wind energy applications and has been discussed in detail by numerous authors [5–7], or by the measurement standards of organizations such as the American Wind Energy Association [8].

Although pressure and temperature are sometimes also measured in the wind farm to compute the air density (wind power is proportional to air density [9]), our study is focused on anemometers and wind vanes that measure wind speed and direction. The wind speed can therefore be decomposed into the zonal and meridional components ($U$, $V$; see the table of abbreviations before the References for the other parameters).

Wind-measuring instruments can be classified according to their principle of operation:

- Momentum transfer: cups, propellers, and pressure plates;
- Pressure on stationary sensors: Pitot tubes and drag spheres;
- Heat transfer: hot wires and hot films;
- Doppler effects: acoustics and laser;
- Ultrasonic devices, etc.

In recent years, it has even been possible to develop a resource assessment study using modern anemometers such as LiDAR (Doppler-effect-type laser-based anemometer) obtaining wind data at different heights at a given location [10]. SODAR (sonic detection and ranging) and meteorological masts can also be used [11], by means of on-site anemometry observation [12], and single anemometers can be used in complex terrain for resource assessment purposes [13].

In our case, cup-anemometers located at the turbines of a wind farm were analyzed, and these are the most common instruments. Baseer et al. recently studied the performance of cup anemometers installed at different mast heights [14], calculating the annual mean, median, and standard deviation. These indicators were almost the same during different years and were comparable with co-located sensors at each height. Thus, the similarity of the measurements of different cup anemometers at the same location (or almost the same location) seems to be demonstrated for modern cup anemometers.

The rotation of a cup anemometer varies in proportion to the wind speed to generate a signal. It presents an extremely linear calibration, but it can start from a zero rotation rate at zero wind to one corresponding to a sudden change. The bias in the measured mean wind speed due to the random variations in the three velocity components is overwhelmingly dominated by the fluctuations of the lateral wind velocity component [15].

The relevance of the lateral wind fluctuations in the measurement of the wind speed that determines the power curve means that wind direction cannot be ignored for the construction of the actual curve. Consequently, an evaluation method that takes into account both wind speed and wind direction (or $U$ and $V$, zonal and meridional components) is necessary, and the verification cannot be reduced to wind speed alone. The stability of the atmosphere and the subsequent turbulent fluctuations is so important that a more stable atmosphere with the same average wind speed produces more energy in a wind farm [16].

In fact, although it is not the aim of this study, the bias due to lateral wind velocity fluctuations can be reduced to less than 1% by means of a special data processing of the simultaneous signals from a cup anemometer and a wind vane [15]. Bias or root mean squared error ($RMSE$) values between the anemometers of the wind farm of around 1% will be therefore referential for our purpose.

For instance, an extreme relative bias of 10% in the measurement of the anemometer can produce important deviations in the actual power curve of the turbine. Figure 1 shows a typical power curve of a wind turbine with the cut-in wind speed around 3 m/s at which the turbine starts its production and the cut-off at 25 m/s at which the turbine stops because of safety issues. The variations for these limits are important, but the changes in the $U^3$ zone of the curve before reaching the rated wind speed with constant power (rated power) are also very important. This is because this kind of deformation of the power curve in the $U^3$ zone is similar for other kinds of technical problems, such as the pitch misalignment of turbine blades, yaw misalignment, or instantaneous turbulent variations of the wind due to the atmospheric instability [16–18].

Recent works of the authors show real cases of energy production diminution due to pitch misalignment in wind farms and simulations with FAST that compute the fraction of power reduction at each wind speed in the $U^3$ zone for different values of pitch errors. These errors can be corrected after an in situ measurement of the blade-hub configuration via laser scanner [17,19,20].

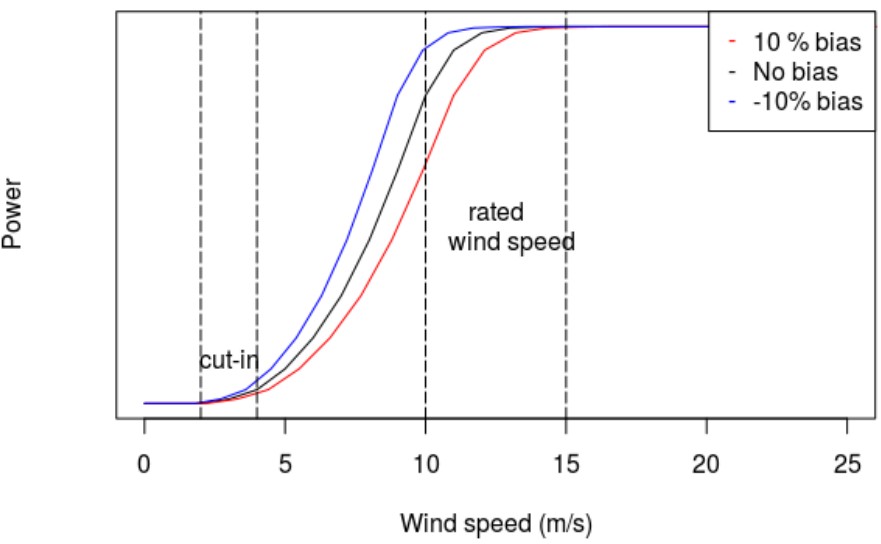

**Figure 1.** Power curves for different biases.

Similarly, analogous deformations of the power curve due to the errors in the yaw angle of the turbine can be described [18]. Consequently, the effects of the pitch and yaw angle deviations in the actual power curve each have their own characteristics, but they may also appear in combination with the effects of the errors in the measurement of the anemometers. That is why the identification of defective anemometers is so important for the wind energy industry.

Over time, cup anemometers' loss of performance due to aging processes can be considerable, and it can even affect the annual energy production (*AEP*) estimations [21]. Therefore, anemometer calibration is very important [22]. Recently, new analytical procedures based on Fourier analysis and aerodynamics have been developed to predict the degradation level of on-field anemometers [23]. This methodology might represent an alternative to the classic approaches used in the present standards of practice such as IEC 64000-12.

In this paper, we want to present the results of a joint approach for the identification of defective anemometers that has already been applied to a real-life wind farm in Northern Europe, and this paper represents the formal presentation of a fully-structured methodology to identify defective anemometers in any wind farm. The multi-criteria methodology we developed involves the combination of

several techniques using a standard reference for comparison (benchmarking) that converge into the identification of the defective anemometers in a wind farm. This approach is not site-dependent and can be generalized to any wind farm in any location with any number of defective anemometers.

## 2. Data and Methodology

In this section, the source of data used for this study and the methodology developed are explained.

### *2.1. Data*

#### 2.1.1. ERA5 Reanalysis

The wind speed data at the nearest ERA5 grid point were used to assess the quality of the wind measurements at the 10 anemometers. This grid point is at a distance between 4 and 8 km from the turbines of the wind farm; without forgetting that the concept "nearest" is complex because it works in a Gaussian grid. The European Centre for Medium-Range Weather Forecasts published this advanced reanalysis in 2017, improving previous reanalyses such as ERA-Interim. The representation of the troposphere and tropical cyclones is better, as is the soil moisture, the balance of precipitation and evaporation, and the consistency between sea surface temperature and sea ice. The data assimilation system is also renewed (IFSCycle 41r2 4D-Var), and a vast amount of historical observations (satellite or in situ) are assimilated [24].

ERA5 provides hourly estimates of a large number of atmospheric, land, and oceanic climate variables. It presents a 30-km global grid and uses 137 vertical levels from the surface up to a height of 80 km. The entire dataset from 1950–present will be available soon, substantially extending the period of ERA-Interim (1979–present), with a much higher spatial and temporal resolution [24]. Furthermore, recent studies in real wind farms have shown that ERA5 can be the new "champion" of wind energy modeling [25]. ERA5's *U* and *V* wind speed components were obtained at 10 and 100 m in the nearest grid point. In order to compute the wind speed at 137 m (the anemometer's height), we first obtained the roughness of the terrain ($z_0$) from the log law using the average values of the wind speed module **U** at 10 and 100 m:

$$\mathbf{U}(z = 100)/\mathbf{U}(z = 10) = log(100/z_0)/log(10/z_0) \tag{1}$$

Therefore, the value of $z_0$ was around 500–1000 mm. According to Table 2.2 in [26], this type of terrain corresponds to forests and woodlands, which is totally consistent with the terrain of the wind farm (not mentioned for commercial reasons). There are better, but more elaborated techniques to perform the vertical extrapolation of wind data [27]. However, considering that in this study, the gridded wind data from ERA5 that is vertically extrapolated is used for relative estimations of errors with different anemometers, its use is deemed as not necessary. On the one hand, because there are other errors such as the ones derived from the representativeness error: comparison of in situ anemometer data to numerical model output at a single grid point with a spatial resolution close to 30 km; or, on the other hand, the spatial distance from the near-neighbor grid point to the anemometers in the wind farm.

Again using the log law with this value of $z_0$, wind speed can be raised from 10–137 m in height. Thus, the mean ratio between $\mathbf{U}(z = 137)$ and $\mathbf{U}(z = 10)$ was 1.8. That is, the wind speed increased 80% from a 10-m height to the hub height of the turbines.

#### 2.1.2. Wind Farm

For this work, measurements obtained from 10 anemometers at the 10-turbine hub of a wind farm located in Northern Europe were analyzed. The anemometers measure both wind speed and direction. This wind farm is located in a flat area of big forests without additional effects such as land–sea interactions or mountain breezes affecting the wind field. In this paper, the 10 turbines in the wind farm are labeled as: WTG-08, WTG-09, ..., WTG-17. The period of study ran from 1 November

2016–31 August 2017 with 10-min data gathered at the 10 anemometers. Therefore, there are 43,776 cases, but not all of them are complete (there are default data or NaNs (not a number)), and the 1.9% of missing data were removed. Therefore, finally, 42,936 complete cases were considered. This is done for all the sub-techniques of MIDAS, with the exception of running plots (see Section 2.2.4), in which a temporal window of one week was used to run the time series.

The ten anemometers analyzed were located at the mentioned height (137 m, turbines' hub), and there were two clear parallel lines of turbines: WTG08, WTG09, WTG10 and WTG11, WTG12, WTG13. The other four were a group somewhat apart (Figure 2) aligned around a W–E direction, thus configuring a reasonable layout for the wind farm given that the two prevailing wind directions are NNE–NNW and SW. This layout and the position of the turbines can be seen in Figure 2, together with the ERA5's nearest grid point.

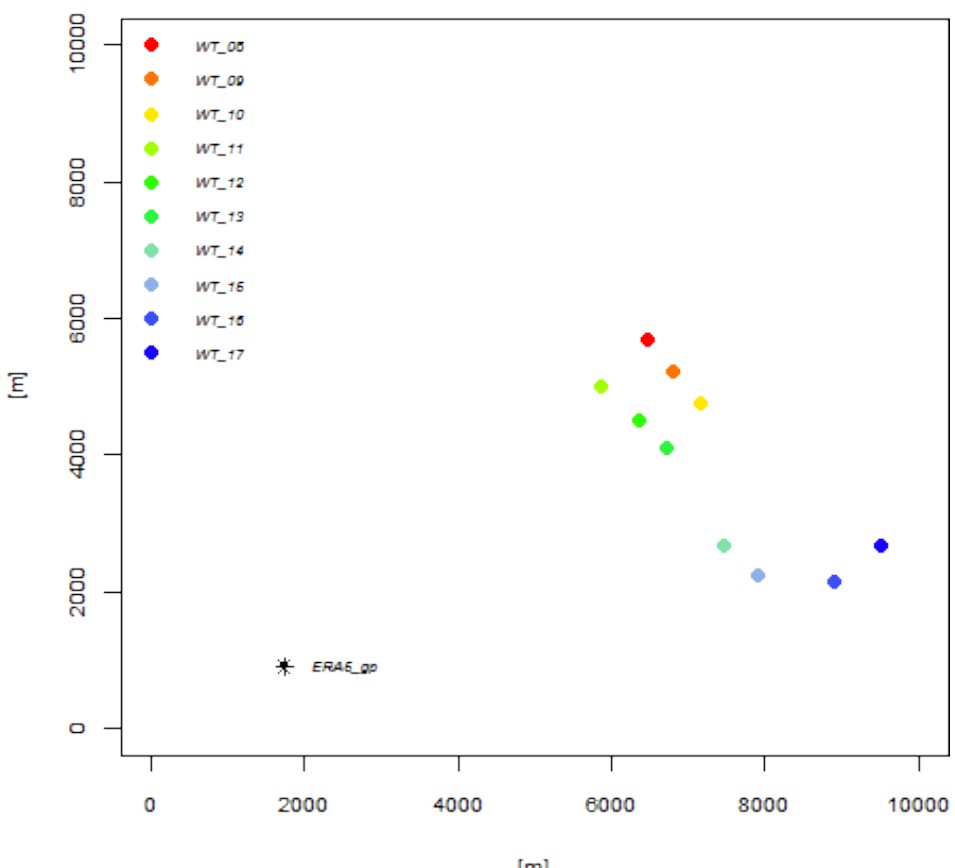

**Figure 2.** Layout of the wind farm and the position of the nearest ERA5 grid point.

Summarizing, the wind data measurements from the anemometers were gathered as speed + direction records every ten minutes, while ERA5 provided the hourly zonal (*U*) and meridional (*V*) wind projections. An initial lengthy phase of data arrangement for compatibility between both wind sources was needed. In it, the following tasks were carried out:

1.  Wind speed + direction measurements from the 10 anemometers were transformed into *U-V* values
2.  At the anemometers, values were taken every ten minutes, while ERA5 records were available only at hourly steps. To arrange both groups of data (ERA5, 10 anemometers) on the same timeline, only 1 in 6 measurements from the anemometers were considered.

3.   Finally, ERA5 only provided wind *U-V* values at a height of 10 and 100 m above the terrain and not at 137 m (hub/anemometer height), so following the log law, ERA5 *U-V* values at 137 m were derived.

*2.2. Methodology*

MIDAS (multi-technique identification of defective anemometers) involves combining five different approaches: multidimensional probability density function estimators, analysis of wind roses, Taylor diagrams, the time series plots for the running correlation, running *RMSE* and running bias, and finally, principal component analysis. These techniques are explained in detail in the following sections.

2.2.1. Multidimensional Probability Density Function Estimator

The multidimensional probability density function (MPDF) technique is a general-purpose method implemented by the authors [28] that makes it possible to compare two multidimensional probability density functions (PDFs) that were estimated by a kernel-based multivariate approach. This method provides a score between 0 (completely different PDFs) and 1 (perfect match) for multidimensional data distributions by computing the common volume under both PDFs.

In general, for the verification of models, it is customary to have a time series of measurements (approximately error-free) and model results, the data that need to be verified. However, in this case, either one (or two) of the components of wind measured by some (or none) of the wind-measuring devices might be affected by observational problems. This is a problem where the true observed state is not well known.

As such, it is similar to the problem of evaluating climate models (models that are always run beyond the first-kind predictability limit associated with weather forecasting) against observations. The climate models cannot be expected to reproduce the daily atmospheric states, but they can be expected to generate a similar probability density function [29,30]. Recently, some of the authors of this study proposed an extension of the previous evaluation strategy to multiple dimensions [28].

This extension to multiple dimensions has many advantages in the present application:

1.   First, it allows us to evaluate simultaneously the probability density function of different components of the wind from different wind-measuring devices, since it supports the analysis of multiple dimensions.
2.   Second, despite the fact that well-known statistical tests exist for the comparison of observed and simulated probability distribution functions (e.g., the Kolmogorov–Smirnov test or comparison of Weibull distributions [31]), it is also well known that the univariate Kolmogorov–Smirnov test cannot be easily extended to several dimensions above two or even three [32,33]. The use of a kernel-based multidimensional probability density function is a more general strategy that can be used very flexibly with a higher number of dimensions.
3.   Finally, the use of a very efficient algorithm [28] makes it also computationally feasible.

To the best of our knowledge, this is the first time that the MPDF technique has been applied for wind field comparison purposes. It allows the vectorial comparison of the wind measurements at two locations and/or obtained by two different devices or from two sources. Since a wind vector is defined by its two components (*U*, *V*), in this work, the MPDF will be applied two-dimensionally.

At the beginning of this study, no information was available as to the number of faulty anemometers and which ones (if any) they were. For this reason, since we could not trust any of the measuring devices at a given moment, an external reference was adopted: the nearest grid point of ERA5. We decided to assume that wind-measuring devices should ideally be representing a similar probability density function unless any of them were affected by some kind of malfunction. To evaluate this, the MPDF was used to compare the 10 anemometers against ERA5, that is we created 11 two-dimensional PDFs corresponding to the wind vector (*U*, *V*).

The practical implementation of this MPDF technique as described in [28] was carried out in three steps:

1. First, the optimal bandwidth that must be used in the estimation of the MPDF was derived by means of smoothed bootstrap, since the Epanechnikov kernel used does not naturally lead to a simple solution by means of cross-validation.
2. Second, for the $U$ and $V$ components of wind-measuring devices, the corresponding two-dimensional probability density function was computed.
3. Finally, the common volume under both multidimensional PDFs was computed for the combination of wind-measuring devices.

The software was written in ANSI-C and is freely available as indicated in [28] (https://github.com/isg-ehu/unai.lopez/tree/master/density-parallel), where more specific mathematical details on every step can be found. In our case, the MPDF is bi-dimensional, and the probability can be visualized in a color-plot ($U$ and $V$ in the $x$ and $y$ axes and the probability in colors). Figure 3 shows the advantage of analyzing the problem from the point of view of a multidimensional probability density function. Both the ERA5 and in situ wind data show a probability function characterized by two local maximums. It can be seen that the different positions of these maxima would project over similar position if only the $U$ component of wind were used. Performing the analysis of the probability density functions (and their match) in two dimensions allows better discriminating the match between model (reference, ERA5) and measured data at each anemometer.

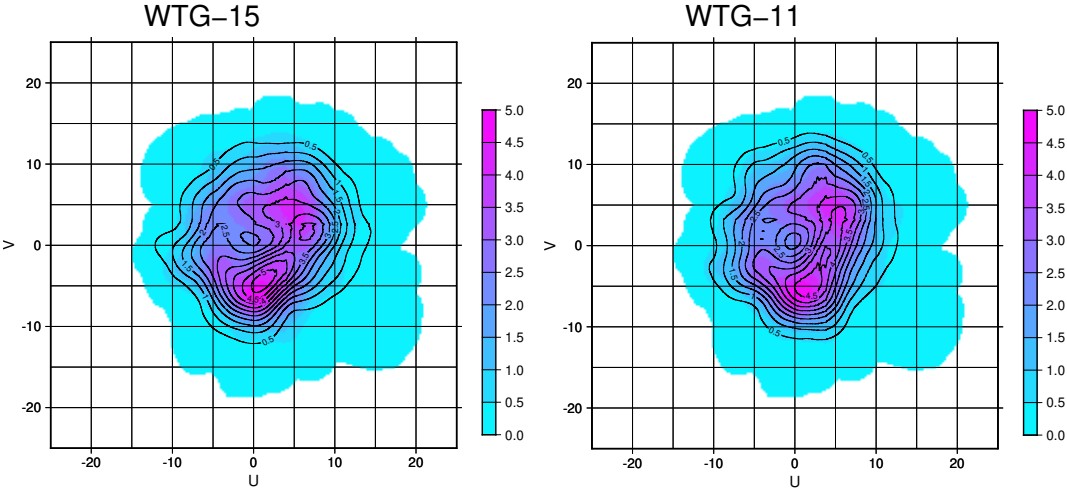

**Figure 3.** Two-dimensional probability density function ($\times 1000$) corresponding to wind from the nearest ERA5 grid point (color shades) and the WTG-15 anemometer and wind-vane (left, contours). The right panel represents the two-dimensional probability density function ($\times 1000$) corresponding to ERA5 data (colored shades) and the WTG-11 anemometer (contours).

As mentioned above, this score ranged from 0–1, and its values were used to establish a ranking of anemometers in the wind farm and how they behaved in comparison with the ERA5 wind field at the nearest grid point.

### 2.2.2. Wind Roses

A first basic visualization of wind data in each turbine can be also obtained using wind rose diagrams. This allows a fast interpretation of the predominant wind direction in the wind farm and of the farm's configuration. The diagonal disposition should be perpendicular to predominant winds in order to minimize the wake effect between turbines. In this way, we have a first visual idea about the consistency between the wind data and the wind farm configuration, and about the results obtained by the MPDF score comparing the wind roses.

### 2.2.3. Taylor Diagrams

The validation of the anemometers against the group of faultless anemometers was also represented using Taylor diagrams [34]. In this way, we passed from a reference at meso-scale level (ERA5) to an in-field reference in the wind farm at a micro-scale level. Hypothetically, all the suitable anemometers should present a very similar behavior, and the results of the statistical indicators of faulty anemometers evaluated against them should also be very similar.

Three statistical indicators are represented in these kinds of diagrams:

1. Root mean squared error ($RMSE$), represented by the arcs with the center in the observation point;
2. Pearson correlation coefficient, represented by the exterior arc; and
3. The ratio of standard deviations between the model and the observation ($SD$ratio), represented by the interior arc in the case of $SD$ratio = 1.

The trigonometric relation that exists between the three statistical indicators (i.e., the cosine law) allows this representation in a single diagram, and in our case, if all the anemometers of the wind farm were working properly, a compact cluster of points would be shown in the diagram. Therefore, any deviation of a point from this cluster would indicate a faulty anemometer.

### 2.2.4. Running Correlation, Running $RMSE$, and Running Bias

Taylor diagrams show three statistical indicators in a single overview. However, the temporal behavior of the indicators cannot be appreciated in this way. For this, running correlations, $RMSE$, and bias of $U$ and $V$ against the group of faultless anemometers were plotted along the period of study with a temporal window of one week. In these representations, we can find the moments when an anemometer starts to fail because of diverse problems that can be related to O&M issues of the wind farm. Both $RMSE$ and correlation indicate absolute values, but the bias can be both positive and negative and shows the deviation of the signal with respect to the reference value in terms of under- or over-estimation.

### 2.2.5. Principal Component Analysis

The study of PCs (principal components) is another common technique in time series that will be applied here. Principal components are defined as linear combinations of the original variables that explain the highest possible amount of variance with the least number of variates. The principal components are always ordered in decreasing order of explained variance so that the most common variability existing in the original dataset can be found in the first principal component (or empirical orthogonal function (EOF), as it is commonly referred to in geophysics). In this way, the computation of the first, second, and subsequent EOFs shows the contribution to the variability of the signal by each component. Under the assumption that the wind field over the farm is relatively uniform (no large spatial asymmetries are expected), the true wind field can be expected to be captured by the leading EOF, while the errors will contribute to the secondary EOFs. In order to improve the readability of results, instead of using the common orthonormal scaling of EOFs, we scaled EOFs (the loading factors affecting every anemometer in the farm) so that they represent the amount of variance at that anemometer that can be explained by each principal component. Thus, if the $i$th principal component allows us to describe the variability of a given $j^{textth}$ anemometer with a high fraction of variance, this means that it is particularly representing the behavior of the $j^{textth}$ anemometer. Under the previously-mentioned assumption that the wind field is spatially uniform, all anemometers should represent the same variability, and the leading EOF should be the only one expected. If a given anemometer projects significantly onto the second EOF, we can interpret this result as being due to the fact that this anemometer is not showing the same kind of variability that is common to the rest of anemometers either because of an observational error or due to the spatial anisotropy of the wind field. In order to test the stationarity of this result, the PCs were computed not only for the whole period, but for monthly subsets, as well.

## 3. Results

### 3.1. Kernel-Based Bi-Dimensional PDF Estimator

Using this tool, the statistical distributions of the ten anemometers were compared with ERA5. A score between zero and one was obtained, indicating the similarity between the distributions of the anemometers and ERA5's wind speed distribution.

First, a common range for the *U* and *V* time series should be established for the construction of the bi-dimensional PDFs taking the minimum among the minima and the maximum among the maxima. In this case, *U* was between $[-22.49, 16.18]$ and *V* between $[-17.04, 17.29]$ m/s.

Figure 4 shows the final scores in a bar plot. All the scores were between 0.90 and 0.92, with the exception of WTG-15, which dropped to 0.88. These results preliminarily classify the anemometers between faulty ones and faultless ones; a classification that must be corroborated with the following multi-criteria steps based on different mathematical techniques. In this case, there was only one defective anemometer, which was called the worst-in-class (WIC). The others showed a similar score against ERA5, and in principle, they constituted the group of faultless anemometers that can be used as a reference in the following steps. In any case, the following diagnosis process must demonstrate this classificatory hypothesis.

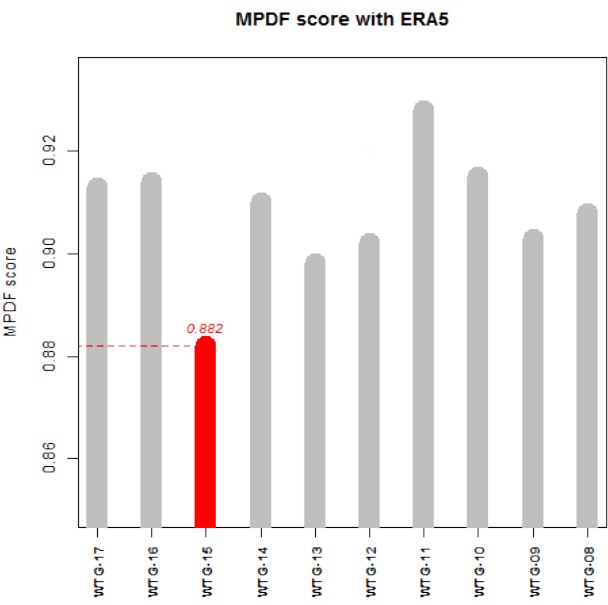

**Figure 4.** Results of the multidimensional probability density function (MPDF) score of the ten anemometers versus ERA5.

### 3.2. Wind Roses

In Figure 5, the wind roses of ERA5 (a), the WIC (b), and a faultless anemometer are shown (c). The other anemometers are not shown because the wind roses were very similar. The same color bar scale is used for the three wind roses in order to have an equal comparison.

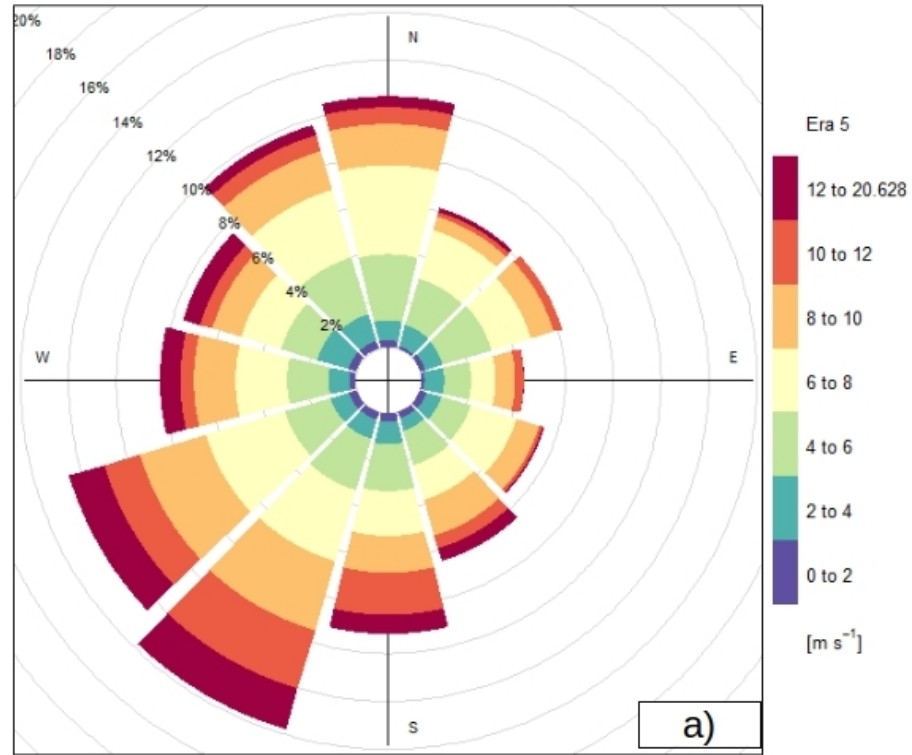

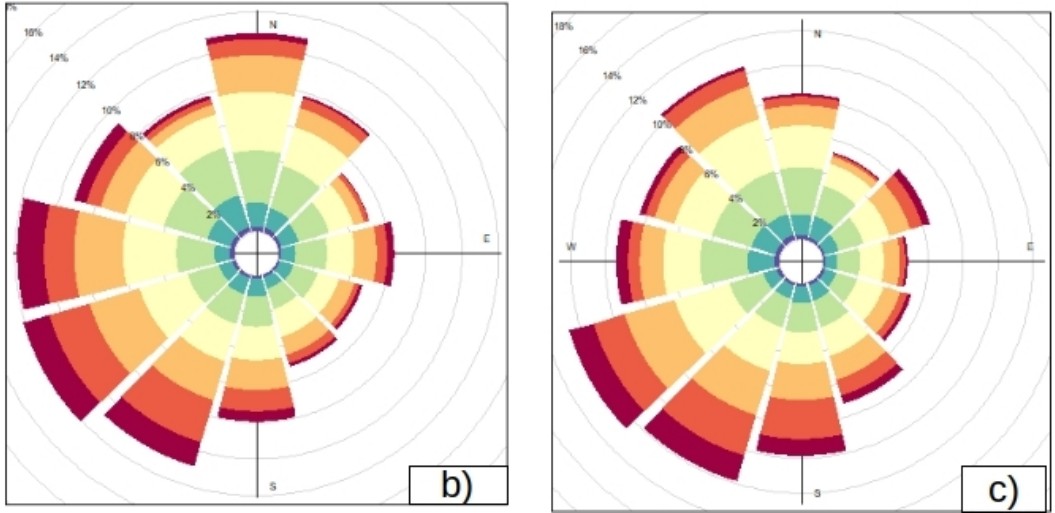

**Figure 5.** (**a**) Wind rose of ERA5's data; (**b**) wind rose of the worst-in-class (WIC) anemometer; (**c**) wind rose of one of the good anemometers (WTG-14).

The predominant wind directions from the SW direction were consistent with the NW–SE diagonal disposition of the farm, since they created perpendicular lines, reducing the space occupation to avoid the wake effect.

There is another very important aspect: it seems that the WIC's wind rose matched with the faultless one and the other anemometers' or ERA5's if it was rotated anticlockwise. Therefore, apart from the deviation of the WIC WTG-15 anemometer, the high similarity between ERA5's wind rose and the other anemometers' wind roses must also be emphasized. The rotation of the WIC's wind rose may mean the presence of an offset in the wind vane, which can affect the turbine in the yaw orientation.

Thus, we could have a case of vane misalignment that should be studied with the following techniques, mainly PCA analysis.

### 3.3. Taylor Diagrams

In Figure 6, the Taylor diagrams are shown for the zonal and meridional components, taking WTG-14 as the reference. The diagrams versus the other faultless anemometers were very similar and are not shown here.

As the diagrams show, the WIC anemometer WTG-15 (represented by the number 7) again had a lower correlation (0.90) and a higher *RMSE* (around 3 m/s) than the other turbines (the group of the other anemometers had a correlation of 0.99 and an *RMSE* of 1 m/s). These diagrams very clearly represent the wrong behavior that was previously identified by our kernel-based bi-dimensional density estimator and show that the data measured by this anemometer were not reliable for both *U* and *V* components.

Furthermore, the group of anemometers considered faultless in the analysis by the MPDF score again showed a very similar behavior, reinforcing the first hypothesis about their suitable behavior.

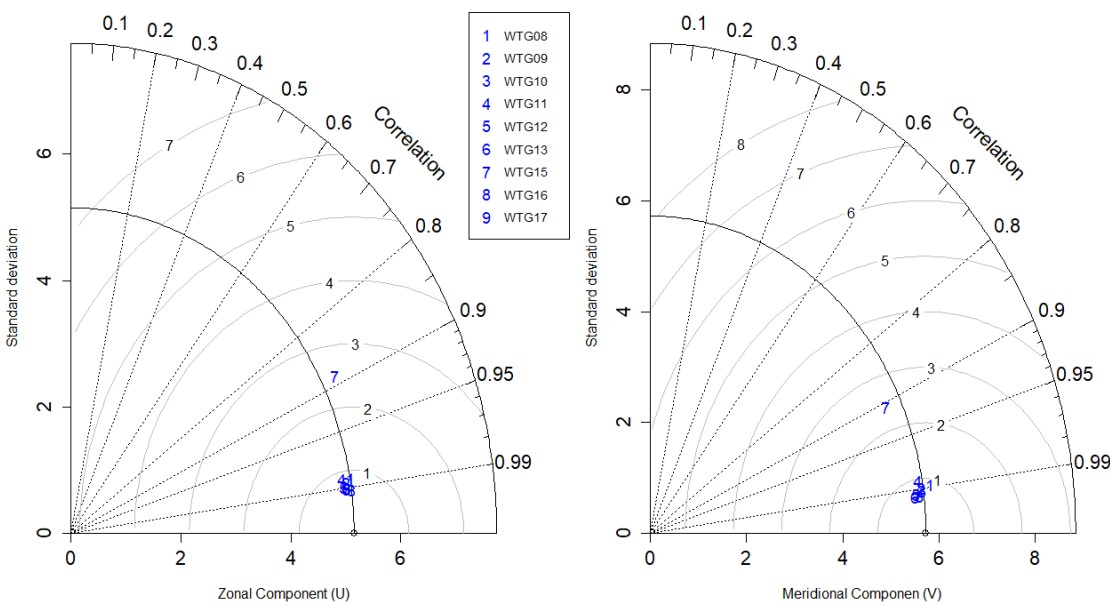

**Figure 6.** Taylor diagrams taking one of the faultless anemometers as the reference: (**left**) zonal component; (**right**) meridional component.

### 3.4. Running Statistical Indicators

In this step, we wanted to reconfirm the obtained results both in the Taylor diagram and in the MPDF score. This step is interesting mainly in its ability to identify the time frame in which the anemometer began to fail and started measuring differently from the rest.

By means of these graphs, the moments when the anemometer of WTG-15 turbine behaved erroneously were identified. To that purpose, time series plots are shown representing the values of the correlation, *RMSE*, and bias in period windows of seven days for *U* and *V*. The same colors were used for the anemometers in the six time series graphs.

Like in the Taylor diagrams, a faultless anemometer (WTG14)was chosen as the reference. We cannot show all the plots against all the faultless ones, but they were very similar. As can be seen in Figure 7, the problems of the WIC anemometer took place all along the year, reaching correlations

below 0.7 and *RMSE* between 2 and 4 m/s, as well as similar positive bias for *V* (upper-estimation) and negative for *U* (underestimation). Given that the mean values of the anemometers were around 7 m/s, this means that in some moments the relative error could be above the 50% for our WIC anemometer (brown color). The others' *RMSE*s were below 1 m/s, and the biases also moved between −1 and 1 m/s, which is an important difference compared with the WIC. In general, the worst results for the three statistical indicators were shown between 1 November 2016 and 13 January 2017, at the beginning of the time series.

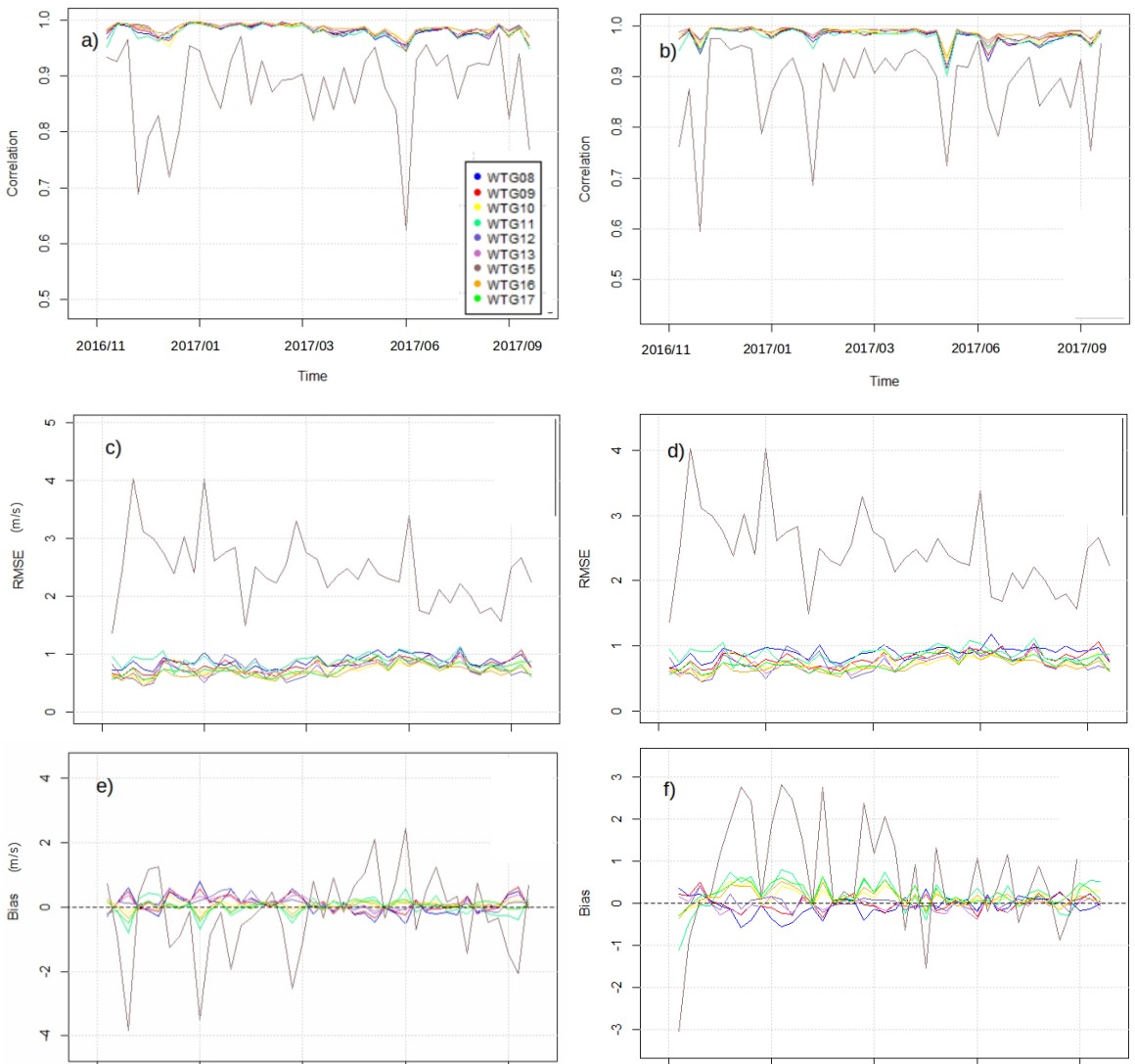

**Figure 7.** Running correlation of: (**a**) zonal component and (**b**) meridional component. Running *RMSE* of: (**c**) zonal component and (**d**) meridional component. Running bias of: (**e**) zonal component and (**f**) meridional component.

*3.5. Principal Component Analysis*

Finally, Figure 8 shows the results from the principal component analysis of zonal and meridional wind speeds. It clearly shows that the results by the MPDF scores and the Taylor diagrams were robust. This can be said because the leading principal component explained the majority of the variance at every anemometer/vane, with the clear exception of WTG-15 (the one already identified as the WIC in our dataset by the previous techniques). The results from WTG-15 were the ones appearing as the most important in the second PC. This again shows that WTG-15 was the anemometer-vane showing

the most different behavior. The leading EOF for both the zonal and meridional components explained 97% of the whole variabilities of both zonal and meridional wind components, with only 2–2.5% for the second principal component, most of it concentrated in the series measured by WTG-15.

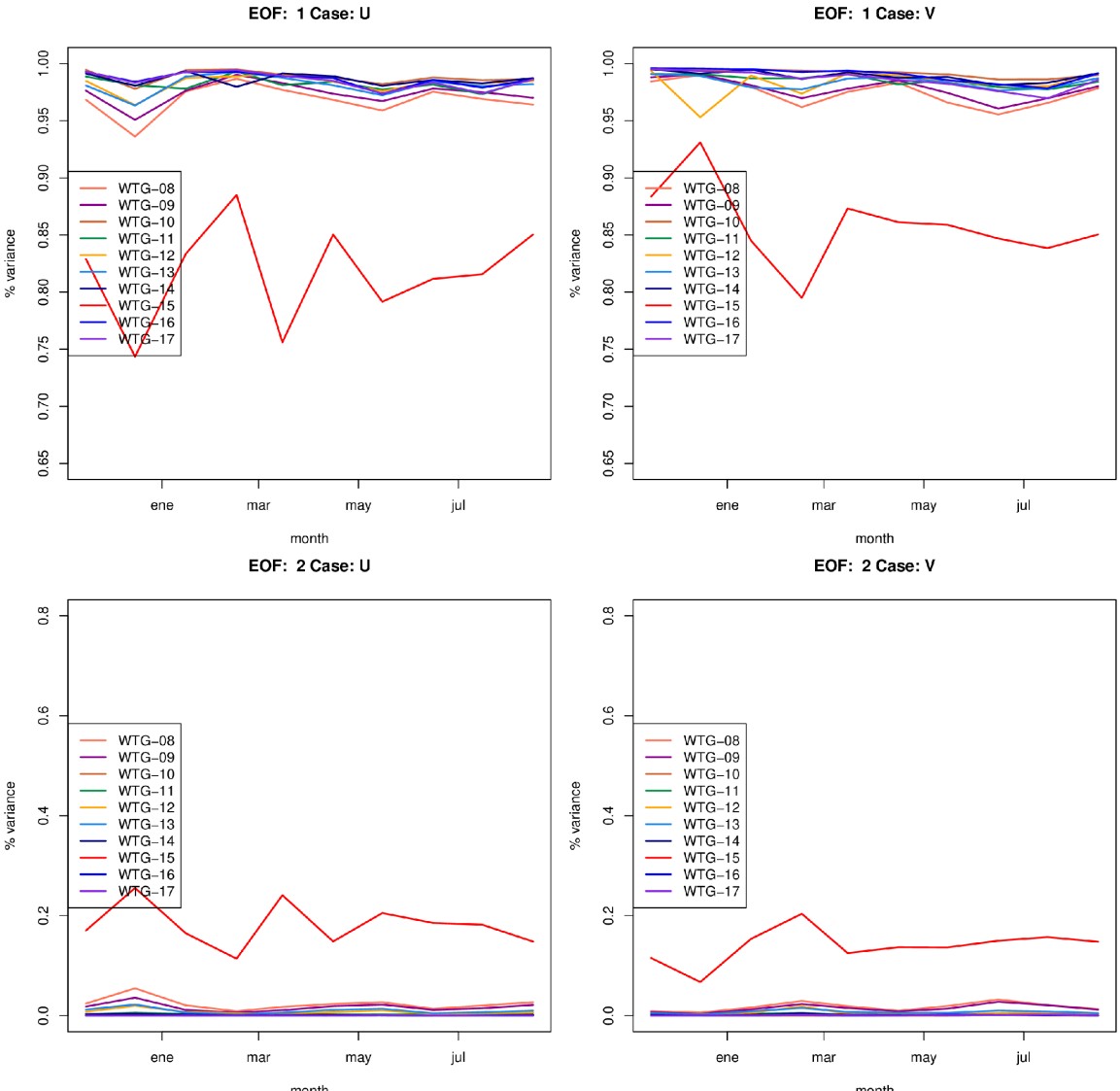

**Figure 8.** Principal components (leading, top row, and second, bottom row) of the zonal (left column) and meridional (right column) wind components for the anemometers in the wind farm, expressed as the fraction of variance explained at every anemometer by each principal component. EOF: empirical orthogonal function.

Thus, the variability measured by WTG-15 was to some extent decoupled from the variability measured by the rest of the anemometers. Finally, if the previous computations (based on the principal component analysis) were repeated for the magnitude of the wind speed, the leading principal component explained 96% of the variance, and WTG-15 did not appear as an outlier and faulty in the distribution of anemometers. Thus, the difference of WTG-15 with respect to the rest of the anemometers points to a stationary misalignment between them.

Since many of the results indicated that the WIC anemometer was probably affected by a misalignment, Figure 9 shows the results of explained variance corresponding to the first principal component that were obtained when a varying rotation angle was applied to the WIC anemometer for both wind components. If both velocity components of the WIC anemometer were rotated

counter-clockwise by an angle of 26°, the leading principal component (representative of the true wind field) explained as much as 99% of the total variance. This confirms that the misalignment detected by comparing the wind roses corresponded to a value of 26°.

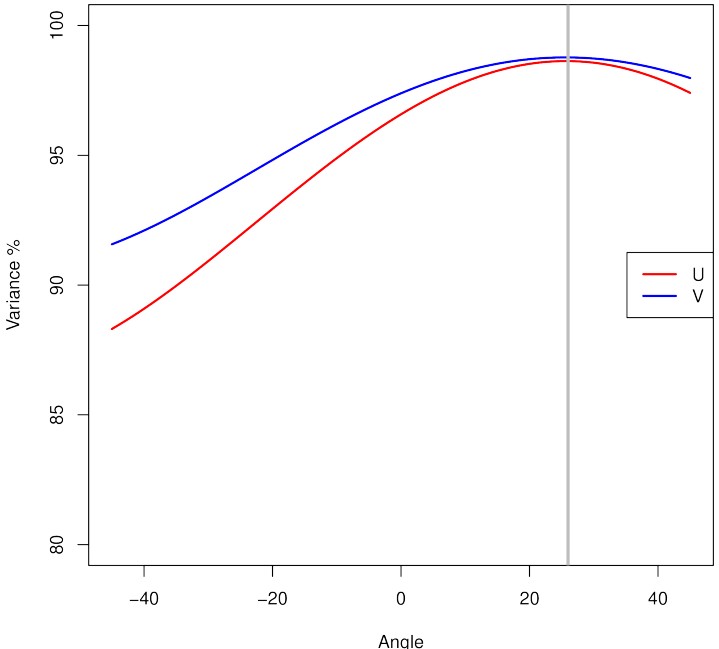

**Figure 9.** Variance corresponding to the leading principal component after a rotation was applied to the wind field components from the WIC anemometer before the principal component analysis (PCA) was computed. The maximum amount of variance in PC1 was obtained with an angle of 26°.

## 4. Discussion

In this study, we used a relatively simple formulation to take into account the vertical dependence of wind with height above the surface. Previous studies [27] have extended the surface-layer theory based on the Monin–Obukhov scaling at higher heights by using different lengths that consider in detail the stability of the atmosphere, derived from measurements in Denmark and Germany. Since the implementation of this scaling depends on the existence of atmospheric measurements at different levels, which allow on to take into account the atmospheric stability, we have preferred to keep the algorithm simple, by vertically scaling the wind from the ERA5 nearest grid point by means of a simple log law. The following considerations support our selection. First, it makes the algorithm simple to be applied to wind farms where a meteorological measurement system at multiple heights does not exist. Second, the extrapolation of wind to different vertical levels is not the only error that exists. On the one hand, wind data from ERA5 correspond to estimations of wind at a grid point representative of the characteristics of wind over a grid cell around 31 km by 31 km and hourly temporal resolution, while wind-farm observational data correspond to observations every ten minutes. Thus, there exist substantial representativity errors that are very likely higher than the error due to the vertical extrapolation using a simple log-law. Besides this representativeness problem, a second factor to consider is that there exist some kilometers between the grid point to the wind farm (from 4 km–8 km in our case).

After the initial screening provided by the visual analysis of wind roses and by the MPDF score against ERA5, it could be seen that there was one single anemometer (WIC) in the wind farm exhibiting a clearly different behavior when compared with the rest. At a second step, one of the other anemometers was chosen as a reference, and the Taylor diagrams clearly indicated a cluster of anemometers seeming the same while the position of the WIC was distant from the rest in both directional Taylor diagrams. The running diagrams corroborated the same observation, adding

information about the time periods with the worst behavior of the WIC anemometer that can be valuable for O&M analysis of the wind farm.

The very defective results of the WIC anemometer (WTG-15) compared to the others could not be explained by its position in the wind farm. Furthermore, the WIC was one of the most isolated turbines in the wind farm, being out of the wake effects of the other turbines. In addition, the chosen reference WTG-14 was the nearest turbine, and consequently, there were no qualitative reasons that could explain its bad behavior in terms of micro-scale effects (e.g., terrain obstacles or turbine wakes).

The weekly relative errors in bias and *RMSE* of 50% in some cases within the temporal series emphasized the need for the adequate measurement of the wind direction if we considered the importance of lateral wind fluctuations [15]. As mentioned before, these errors can be reduced to 1% by using special data processing that combines the cup anemometer and wind vane data.

The MIDAS methodology was applied to this wind farm and is a combination of five different approaches: multidimensional probability density function estimators, analysis of wind roses, Taylor diagrams, calculation of the running correlation, running *RMSE* and running bias, and finally, principal component analysis.

This is not our case, but if the met mast is included in the SCADA data of the wind farm, it should be an important reference to develop the Taylor diagrams and the running plots. Therefore, it should be added in the cluster of wind turbine anemometers and in the initial comparison with ERA5.

While the analysis of wind roses and Taylor diagrams provided a preliminary identification of the faulty anemometer, the MPDF estimator allowed an accurate evaluation of the differences in the measured wind components. The principal component analysis made it possible to identify that the error was not in the wind vector module, but in the direction. Therefore, in this case study, a vane misalignment is the cause of the fault. Although an offset in the wind direction measurements can be detected with other numerical and experimental techniques [35], PCA offers valuable and coherent additional information in the general step-by-step perspective of the MIDAS analysis. Finally, the analysis of the running indicators made it possible to identify the exact point in time in which the faulty anemometer started to fail.

The combined approach of these methodologies allowed an in-depth identification and characterization of the faulty anemometer for this wind farm. Although developed for this specific wind farm, our methodology can hopefully be applied to any wind farm in which any number of faulty anemometers may be operating. This must in any case be tested with further studies for different wind farms with, possibly, larger spatial anisotropies in the wind field. The results in other wind farms may contribute more challenging case studies to improve some specific aspects of this general-purpose methodology, but in our view, MIDAS represents a solid methodology capable of providing an accurate diagnosis of the nature of the error (in this case, wind direction), and when those errors started to take place.

Although most wind farm locations are specifically designed to have a wind field that is as homogeneous as possible, in some cases, a few turbines might be deployed close to nearby boundary obstacles that may affect the wind vector. For this reason, this methodology involves an initial stage of an evaluation by experts to adapt this methodological approach to the specific wind farm being analyzed. In this line, although applied in this paper to a specific onshore wind farm, MIDAS could easily be applied to any offshore wind farm where a far more homogeneous wind field can be expected.

## 5. Conclusions and Future Outlook

MIDAS, a multi-criteria diagnosis method with a benchmarking approach for the detection of defective anemometers with different logical steps, was presented in this paper:

1. The first and main step is based on a new multi-dimensional probability density estimator computing a similarity score between the analyzed anemometers and the data offered by also the new ERA5 reanalysis. This allows a first division between defective anemometers and the group of anemometers with suitable behavior.

2.  Having identified the group of suitable anemometers, these can be used as a reference for the application of other statistical and visualization techniques. In this way, we pass from a reference at a meso-scale level (ERA5) to an in situ reference at a micro-scale level in the wind farm.

3.  Although the expected results against all of the faultless anemometers should be very similar, it is convenient to certify this aspect. If so, it will be enough to show the representations against one of the faultless anemometers.

4.  Taylor diagrams and running plots for correlation, *RMSE*, and bias show coherent results compared to the results of the MPDF score. Furthermore, the PCA components also show very robust results.

5.  The feedback of these additional validations can reinforce, as in the studied case, the validity of the first MPDF score against ERA5.

Thus, an integral method for the identification of defective anemometers was developed based on the results given by the MPDF score against ERA5 to identify the faultless and defective anemometers and establish it as the main reference for following well-known unidimensional validations. Definitively, it can be generally considered as a benchmarking method in wind farms, analogous to the method used by the authors for pitch misalignment correction [17,19,20]. Mathematically, MIDAS constitutes a robust and generalist methodology unifying several statistical techniques in a benchmarking approach, but it must be applied in other types of terrains and wind farms before having a relevant and definitive evaluation about its performance in different types of terrains and atmospheric conditions. Future research with a more extensive casuistry is needed to generalize MIDAS, because this is a paradigmatic case to show the general methodology.

In any case, it must be emphasized that, as far as we know and although they are well-known in meteorology, Taylor diagrams are used for the first time in this context of the wind energy industry's O&M, and they show a representative visualization of the deviations of the anemometers in a single diagram that is able to express three statistical indicators.

Although our objective was to present a robust and simplistic method, the results obtained can be added to those obtained using other approaches based on CFD, wake analysis by mesoscale models, or tower shadow effects [36–38].

Additionally, if pressure, temperature, and moisture data are measured in the wind farm, the third variable that defines the wind power density can be introduced in the MPDF score: the air density. This would not be relevant for an evaluation within the wind farm, but it can be used for the verification of mesoscale models or reanalysis (ERA-Interim, ERA5 in an advanced 'meso-beta scale' [39]) in the nearest grid point [24,40]. This would produce a qualitative leap for the validation of wind energy, since this method would consider all the variables in a single score.

Besides, this multi-dimensional validation can be extended to other kinds of renewable energies such as wave energy, in which the power is determined by two variables (wave height and period) and Taylor diagrams are also used for validation of meso-scale models or reanalysis against buoys [41–43].

**Author Contributions:** Conceptualization, A.U., G.I.-B., J.S.; methodology, A.U., G.I.-B., J.S.; software, A.R., A.U., G.I.-B., J.S.; validation, A.R.; investigation, A.R., A.U., G.I.; writing, original draft preparation, A.R., A.U.; writing, review and editing, all the authors; supervision, all the authors; project administration, U.E., A.U.; funding acquisition, U.E., A.U., J.S.

**Funding:** This work was financially supported by the Spanish Government through the MINECO project CGL2016-76561-R (MINECO/ERDF, UE), the University of the Basque Country through the Euskoiker PT10477 and GIU 17/002 contracts, and the project DIANEMOS of the Council of Gipuzkoa with Maxwind-Hispavista. ERA5 data were downloaded at no cost from the MARSserver of the ECMWF. Most of the calculations were carried out in the framework of R [44].

**Conflicts of Interest:** The authors declare no conflict of interest.

## Abbreviations

The following abbreviations are used in this manuscript:

| | |
|---|---|
| MIDAS | multi-technique identification of defective anemometers |
| WIC | worst-in-class anemometer |
| $z_0$ | roughness of the terrain |
| *RMSE* | root mean squared error |
| MPDF | multidimensional probability density function |
| PDF | probability density function |
| *U* | zonal wind speed |
| *V* | meridional wind speed |
| **U** | wind speed's module |
| *SD* | standard deviation |
| O&M | operation and maintenance |
| PCA | principal component analysis |
| EOF | empirical orthogonal function |
| *AEP* | annual energy production |
| SODAR | sonic detection and ranging |
| CFD | computational fluid dynamics |
| LiDAR | light detection and ranging |

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
