# Peer review of "MIDAS: A Benchmarking Multi-Criteria Method for the Identification of Defective Anemometers in Wind Farms"

_energies, doi:10.3390/en12010028_

Round 1

Reviewer 1 Report

The manuscript presents a method of doing data analysis to find cup anemometers that have errors. It is carefully prepared and easy to follow. It is only about data analysis and no new theory is presented, so the paper is more useful when the code is shared. I would recommend the authors to make their code available somewhere. Otherwise some minor comments are given below:

lines 17-19: this is 'over-selling' the method: in complex terrain there can be large terrain induced turnings and accelarations, so this method is likely to work only in non-complex terrain at large heights, where the ERA5 grid point is representative for the flow. Also it should state somewhere in the abstract which type of terrain the wind farm was located (i.e. terrain elevation differences and surface roughness).

lines 124-126: this is not the right way to extrapolate wind speed above 100 m: above the surface layer the wind does not follow the logarithmic law, see e.g. Gryning, S.-E., Batchvarova, E., Brümmer, B., Jørgensen, H., & Larsen, S. (2007). On the extension of the wind profile over homogeneous terrain beyond the surface boundary layer. Boundary-Layer Meteorol., 124(2), 251–268. http://doi.org/10.1007/s10546-007-9166-9. Furthermore the influence of stability is significant when comparing in the time domain, e.g. in Fig. 7. It would be useful to discuss these topics in more detail here or in the discussion. Differences in stability also mean that this method might not work in coastal areas.

lines 140-141: it is not clear to me if data are removed for one turbine or if a 10-min period is missing at one turbine it is removed in all other turbines as well?

line 220: Mesoscale is usually used for weather models that run at resolutions of 1-10 km grids, so I would remove that word in connection with ERA5.

Fig. 5: green text is difficult to read for the lower panels. Remove or enlarge.

Fig. 6: also here the labels are difficult to read. Please enlarge.

Fig. 7: x-axis labels are too small here. Also please add units to y-axis

Author Response

Attached all the answer to all the reviewers

Reviewer 2 Report

The manuscript entitled “MIDAS: A Benchmarking Multi-Criteria Method for the Identification of Defective Anemometers in Wind Farms” is very interesting and deals with a very important issue in the control and monitoring of wind turbine performances: namely, the power curve of wind turbines can’t be assessed reliably if the anemometers has defects.

The proposed methods are very valuable and original.

I have one main concern and it deals with the selected test case. An offset in the wind direction measurements at wind turbine hub can very likely be detected with the technique shortly described in Castellani, F., Astolfi, D., Piccioni, E., & Terzi, L. (2015). Numerical and experimental methods for wake flow analysis in complex terrain. In Journal of Physics: Conference Series (Vol. 625, No. 1, p. 012042). IOP Publishing at the end of page 5. Therefore, I think that the application of the proposed methods would be much more interesting if the type of defect was more challenging. Therefore, I invite the authors to consider if they have the possibility of presenting a more interesting test case.

As regards the presented test case, I have a question: does the defect in the anemometer produce any anomaly in the power curve plot? Would you consider including some discussion about this in the manuscript?

Author Response

(The authors gave the same response as above.)

Reviewer 3 Report

The authors present a quite extensive and comprehensive research with clear applicability to O&M of large wind farms. The authors believe this methodology to be useful in different terrains, offshore, onshore and complex. However, only one example is presented, and it is my believe that the claims should be toned down along the text until such demonstration is available for the scientific community. 

Line 42 - A principle of operation is missing. How the authors would classify an ultrasonic anemometer/thermometer?

Line 77 - Recommend replace security by safety

Lines 74-81 

    The whole paragraph misses references. Maybe some of the statements are a translation of the authors personal experience but the readers should also have access to the original research where those statements are derived from. 

    Figure 1 is very explicit of the effect of the bias on the power curve. Could the authors reference works where this specific shape of power curve is presented? 

    The phrase “This is because…other kinds…misalignment of turbine blades” is very vague and without references.

Lines 86-90 - Errors in yaw misalignment are known since the 90’s. Please try to state this problem as a known one, not a novelty.

Section 2 - Please make it clear to the reader what data was really used: the ERA5 nearest grid point measured at xx m (agl); 10 data set’s from WT anemometer’s? 

    The wind farm did not have a met tower? If had, why those series where not included?

Figure 3 is unreadable. Some clear information may be obtained and shared to the readers with a surface map and contour lines. The intersection volume is not depicted, therefore the intended effect of presenting these graphs is lost.

Line 361 - Maybe replace “weekly” by “weakly”

Line 369 to 373 - The authors demonstrate clearly that the anemometer was misaligned. Some questions related to this paragraph:

Is not the wind sensor then the responsible when you have a pair cup/wind vane? Please clarify this to the readers;

Is a misaligned anemometer considered a defective one. Was the anemometer misaligned and defected too? The text is not clear in this point.

In line 370 would it be clear to refer to the anemometer, instead of faulty, an outlier? 

Author Response

(The authors gave the same response as above.)

Round 2

Reviewer 1 Report

The revision addresses the problems of the previous manuscript, but some of the new text contains very long lines that are difficult to follow. Please rewrite for example line 131-137.

Author Response

Thanks for the recommendation. We have rewriten the long sentences in the lines you mention and in the discussion. Changes in bold text.

Reviewer 2 Report

The authors recognize the importance of the main point that I have noticed about their work.

The issue practically hasn't been faced in this manuscript, because the authors say that they have problems in obtaining authorization of data publishing from wind energy companies.

At least, the discussion about this point has been improved. 

This is in my opinion sufficient for considering the manuscript acceptable. Anyway, I look forward to read further work by the authors about other applications of the methods proposed in this work.

Author Response

Thanks. We will try to apply the MIDAS method in new wind farms and terrains, and the intention is to obtain the complete SCADA data for that, also the instantaneous power production.

Reviewer 3 Report

Smart changes from the authors. Thank you

Author Response

Thanks for the comments.